Age and growth of chum salmon from the Tumen River

Wang Anqi 1 2 3 4
Li Peilun 1 2 3
Tang Fujiang 1 2 3
Xiong Shuhan 1 2 3
Yu Jiaoyang 1 2 3
Wang Jilong wangjilong@hrfri.ac.cn wjl0321225@163.com 1 2 3
1 Heilongjiang River Fisheries Research Institute, Chinese Academy of Fishery Sciences , Harbin , China
2 Scientific Observing and Experimental Station of Fishery Resources and Environment in Heilongjiang River Basin, Ministry of Agriculture and Rural Affairs , Harbin , China
3 National Agricultural Experimental Station for Fishery Resources and Environment , Fuyuan , China
4 Tianjin Agricultural University , Tianjin , China
Doğdu Servet
Electronic publication date: 2025 Aug 28
Publication date: 2025
Volume: 13
Electronic Location ID: e19902
Received 2025 Mar 10; Accepted 2025 Jul 22
Copyright: ©2025 Wang et al.
Copyright year: 2025
Copyright holder: Wang et al.
License: This is an open access article distributed under the terms of the Creative Commons Attribution License, which permits unrestricted use, distribution, reproduction and adaptation in any medium and for any purpose provided that it is properly attributed. For attribution, the original author(s), title, publication source (PeerJ) and either DOI or URL of the article must be cited.
License URL: https://creativecommons.org/licenses/by/4.0/

Keywords: Tumen River, Chum salmon, Age, Growth characteristics

Funding: The Special Financial Funds of Species Resources Conservation project of the Ministry of Agriculture and Rural Affairs of China (Investigation and Evaluation of Chum Salmon Resources) 2130135 Central Public-interest Scientific Institution Basal Research Fund, CAFS NO.2023TD07 This work was supported by the Special Financial Funds of Species Resources Conservation project of the Ministry of Agriculture and Rural Affairs of China (Investigation and Evaluation of Chum Salmon Resources) (2130135) and Central Public-interest Scientific Institution Basal Research Fund, CAFS (NO. 2023TD07). The funders had no role in study design, data collection and analysis, decision to publish, or preparation of the manuscript.

==============================
To investigate the age structure and growth characteristics of the chum salmon (Oncorhynchus keta) population in the Tumen River, this study conducted age determination and measured basic biological indicators including fork length (FL) and body weight (BW) for 331 specimens collected between 2020 and 2023. Results revealed that the spawning population comprised four age groups (2+, 3+, 4+, and 5+), dominated by age-2+ and age-3+ individuals (96.37% of the total). FL ranged from 51.0 to 74.6 cm and BW ranged from 1,400 to 5,200 g. The relationship between BW and FL was expressed as: BW = 0.0159×FL2.8969, indicating isometric growth. Von Bertalanffy growth equations (VBGF) were applied to examine growth patterns among age groups and between sexes, with parameters estimated via maximum likelihood. Analysis of the residual sum of squares (ARSS) indicated no significant growth differences between sexes but revealed significant differences between the age-2+ and age-3+ groups. The size at maturity (FL50%) for females and males was 45.87 cm and 43.93 cm, respectively, with corresponding ages at maturity (T50%) of 2.00 and 1.87 years, respectively. These findings provide essential biological data for science-based management and conservation of Tumen River chum salmon.

Introduction

Chum salmon (Oncorhynchus keta) are semelparous and anadromous. They spawn in freshwater, and fry outmigrate to the marine environment shortly after emergence. Maturing adults return to their natal streams to spawn at various ages, typically between two and six years. All die after apawning (Salo, 1991). They are widely distributed across the northern Pacific Arctic Oceans (35°N–73°N, 120°E–123°E) and their tributary rivers (Salo, 1991; Urawa et al., 2004; Wang et al., 2011). In China, they are restricted to the Heilongjiang, Ussuri, Suifen, and Tumen River systems (Chen et al., 2005). Historically, Tumen River annual chum salmon catches exceeded 100,000, but spawning ground degradation caused by hydropower development and environmental pollution, combined with overexploitation, reduced annual catches to 2,000–3,000 (Kang & Wan, 2016; Wang et al., 2023). Recent conservation measures, including seasonal fishing bans and annual releases of 400,000–500,000 hatchery-reared juveniles, have stabilized returns at 4,000–6,000 individuals. Notably, hatchery-reared returnees account for only 1.41% of the total (Han et al., 2020; Yan et al., 2021), indicating persistent wild population dominance. The sustained low abundance highlights the urgent need for conservation.

The study of biological characteristics is a prerequisite for fish resource conservation, with the determination of key biological traits such as age and growth being crucial for implementing rational conservation strategies. Through research on age and growth, researchers can not only elucidate growth characteristics, but also analyze growth differences across genders, geographical distinct populations of the same species, and congeneric spscies. Additionally, by fitting growth equations, growth parameters such as growth coefficients, initial growth age, and maximum body length can be obtained, providing basic data for determining their life history types and predicting population dynamics (Zhao et al., 2005; Wang et al., 2013). Research on the age and growth in chum salmon typically uses scales as the analytical material (Morita & Fukuwaka, 2006; Saito et al., 2011; Shimoda, Watanabe & Ando, 2019). Scales annulus formation is related to the fish’s experience of annual water temperature cycles (Wang, Tang & Liu, 2012), and Newman, Cappo & Williams (2000) confirmed that a seasonal change of 6 °C can form one annulus. Combined with the species’s extensive migratory habits in the ocean (Helle & Hoffman, 1995), the seasonal temperature changes it experiences in the marine environment forge clear annuli on its scales. Utilizing the annulus information from scales of returning chum salmon, Wang et al. (2013), Wang, Liu & Tang (2013) and Wang et al. (2020) determined the age composition of populations in the Ussuri River, Suifen River, and Heilongjiang River, finding dominance of younger individuals (specifically 3 and 4 years old fish), and revealing that the group with a younger age of sexual maturity had a faster growth rate. Although sexual maturity in fish is commonly believed to correlate with body size, Morita & Fukuwaka (2006) pointed out that the fish’s prior growth history is a significant factor influencing maturation. Affected by temperature, prey abundance, climate change, and other factors, chum salmon growth exhibits fluctuations (Volobuev, 2000; Kasugai et al., 2012; Kitada & Kishino, 2019). To understand changes in maturation age and body size within chum salmon populations, long-term monitoring of their growth is necessary.

Although abundant biological studies on the age and growth of this species have been conducted internationally (Kaeriyama, 1996; Kaeriyama & Edpalina, 2008; Seo, Kudo & Kaeriyama, 2009), research within China has predominantly concentrated on chum salmon populations in the Ussuri River, Suifen River, and Heilongjiang River basin (Wang et al., 2013; Wang, Liu & Tang, 2013; Wang et al., 2020). To date, no studies have reported the biological characteristics of the Tumen River population. To address this gap, this study examines the Tumen River chum salmon population from 2020 to 2023, with the following objectives: (1) to analyze their age structure using scales as age-determining material. (2) to characterize fork length (FL) and body weight (BW) distributions, back-calculate FL growth utilizing the relationship between scale radius and age, model growth equations (including the growth rate equation), and determine the size at sexual maturity (FL50%) to assess growth characteristics. (3) to compare results with chum salmon populations from the Ussuri River and the Suifen River, clarifying the of age and growth status of the Tumen River population and analyzing potential causes. These findings aim to provide foundational data for conservation and management of chum salmon resources.

Materials and Methods

Sample collection

Study specimens were broodstock from stock enhancement programs, captured near Fangchuan Village (located in lower Tumen River, Fig. 1), using three-layer drift gill nets (mesh size: 12.67 cm). Annual sampling was conducted in mid-October (2020–2023). Following capture, the live fish were transported to a fish hatchery and stocking center for temporary holding. During artificial propagation, biological measurements were performed, and 6–10 dorsal scales above the lateral line were collected and transported to the laboratory for examination and analysis. A total of 331 chum salmon were sampled, including 241 females and 90 males. All sampling procedures strictly adhered to the guidelines provided by Heilongjiang River Fisheries Research Institute of CAFS for Laboratory Animal Welfare and Ethical Review (No.: 20200920-001).

Materials handling and data analysis

Scale processing

Scales were processed following Wang, Tang & Liu (2012), Wang et al. (2013) and Wang et al. (2020). The scales, stored in scale envelopes, were soaked in a 5% sodium hydroxide solution for 5 min, rinsed, dried with filter paper, and mounted on glass slides. Scale annuli were observed under a stereomicroscope (objective lens × 4, eyepiece × 10), and images were captured with a CCD camera for age identification and scale radius measurement.

Age determination

A professional researcher counted annuli on all collected scales. For each fish, the best-preserved scales with clear annuli were selected for analysis due to observed resorption in some samples. Age determination was initially based on the “calendar year method” (Seo et al., 2006), where one year was added to the annulus count (e.g., 2+ corresponding to 3 years old). However, the distance from the scale margin to the last annulus at capture primarily represented the pre-spawning spring-summer period, which did not constitute a full year. To improve accuracy, the “+” value was adjusted to the annual ring value plus 0.7, based on observations that chum salmon hatched from January to February and reached maturity around October (Wang, Tang & Liu, 2012; Wang et al., 2013). Consequently, 2+ and 3+ fish were recorded as 2.7 and 3.7 years old in this study, respectively.

Figure 1 Location of Tumen River chum salmon sampling site (Fangchuan site highlighted).

Fork-length estimation

Scales record growth history through annuli, enabling fork length reconstruction by measuring radial distances from the scale focus to each annulus and margin (Fukuwaka & Kaeriyama, 1997). As back-calculated fork length growth showed significant correlations with observed growth (Fukuwaka & Kaeriyama, 1997), the biological intercept method (Campana, 1990; Francis, 1990; Morita et al., 2005) was applied to estimate historical fork lengths for 331 chum salmon in this study. The formula is defined as: Lt = LC+(St-SC)/(SC-Sb)•(LC-Lb). Where LC and SC are fork length at the time of capture and the total scale length, respectively, Lb and Sb are fork length and scale length at the time of scale formation, Lt and St are fork length and annulus length at the age t. According to the results of previous scholars (Fukuwaka & Morita, 1994; Morita et al., 2005), Lb = 4 cm; Sb = 0.114 mm.

Body weight-fork length relationship and growth equation

The relationship between body weight (BW) and fork length (FL) was evaluated using a power function: BW = aFLb, where a and b were regression coefficients (Ricker, 1975). Analysis of covariance (ANCOVA) was used to detect the differences of the BW-FL relationship between sexes (Fukuwaka & Kaeriyama, 1997; Williams & Monge, 2001).

To quantify maturity age and fork length in chum salmon, we applied the Von Bertalanffy growth function (VBGF) to model growth based on back-calculated fork length data across age groups. The model is expressed as: Lt = L∞⋅[1 − e−k(ti−t0)], where the growth rate is derived as the first derivative of the VBGF: dL/dt = L∞ke−k(ti−t0). The growth performance index (φ) was used to analyze the growth characteristics of chum salmon, defined as: φ = lnk + 2lnL∞ (Pauly & Munro, 1984). Parameters were estimated via maximum likelihood (Wang, Tang & Liu, 2012; Wang et al., 2013; Wang et al., 2020). Here, Lt represents the FL at the age t (years), L∞ is the theoretical maximum FL, k is the growth coefficient, ti is the age (i), t0 is the hypothetical age at FL 0.

To determine the significance of the differences in fork length growth among age groups at sexual maturity, the analysis of the residual sum of squares (ARSS) was used to test for differences (Chen, Jackson & Harvey, 1992). F=RSSP−RSSSDFRSSP−DFRSSSRSSSDFRSSS

where RSSS is the residual sum of squares of different groups of chum salmon growth and other indicator parameters, RSSP is the sum of residual sum of squares of the mixture of groups, DFRSSP DFRSSS is the model degree of freedom, F is the statistic.

Size at sexual maturity

The Tumen River chum salmon population exhibits individual variability in age (2+ to 5+, n = 331) and fork length (FL: 51.0–74.6 cm) at sexual maturity. To standardize maturation thresholds, it is necessary to determine the size and age at maturity, which is defined as the size (FL50%) at which 50% of individuals in a population attain sexual maturity (Zar, 1999), and the corresponding age is termed the population’s sexual maturity age (T50%). The critical maturation phases for salmonids occur in spring (6 months pre-maturation) and autumn (12 months pre-maturation) (Campbell, Dickey & Swanson, 2003; Campbell et al., 2006), synchronized to annulus formation cycles (Morita & Fukuwaka, 2006). Following the study by Morita & Fukuwaka (2006), the onset of sexual maturity is defined by the formation period of the last annulus on scale. For example, in 3+ age individuals, the back-calculated FL at the time at which the third annulus was formed may represent their body sizes at the onset of sexual maturity. Using logistic regression (Anonymous, 1983; Chen & Paloheimo, 1994), we analyzed the probability of maturity within 2-cm FL intervals. The specific formula is as follows: Pi=11+e−a+bli

FL50%=−a/b

where a, b represent the parameters of the formula, i represents the group number, Pi represents the percentage of mature individuals to FL, and li represents the length of each fork of the group.

Age structure

The age composition of Tumen River chum salmon population consisted of four groups: 2+, 3+, 4+, and 5+ (Table 1), with individuals of 2+ and 3+ age groups accounting for 96.37% of the total. Females were dominated by the 3+ age group (51.86% of females), exhibiting a mean age of 3.31 years and a T50% of 2.00 years. Males were predominantly 2+ years old (56.67% of males), with a mean age of 3.14 years and a T50% of 1.87 years. Comparative analysis with the Ussuri River and Suifen River populations (Table 2) showed that both the overall mean age and the female mean age of the Tumen River poupulation were lower than those of the other two populations. Conversely, the mean age of Tumen River males was lower than that of Ussuri River males but higher than that of Suifen River males. Additionally, the T50% values for both sexes within the Tumen River population were lower than those within the Suifen River population. Across all populations, females consistently showed higher mean age and T50% values than males. These results demonstrate distinct age differences among populations and between sexes.

Table 1 Age composition of Tumen River chum salmon (comprising four age groups: age-2+ to age-5+; dominated by age-2+ and age-3+, collectively constituting 96.37% of the total).

Age group	Female	Male	Total	
2+	43.57%	56.67%	47.13%	
3+	51.87%	42.22%	49.24%	
4+	4.15%	1.11%	3.32%	
5+	0.41%	0.00%	0.30%	
Notes.

Note: “+” indicates that the age is the annual ring value plus 0.7, for example, “2+” means that the age is 2.7.

Table 2 Age comparison of chum salmon across populations (mean age and age at sexual maturity (T50%) as comparative indicators; Tumen River population exhibited earlier maturity compared to other populations).

Geographical population	Sex	Mean age	Age at sexual maturity (T50%)	Survey year	Source	
Ussuri River population	Female	3.43	–	2010–2011	Wang et al. (2013)	
Male	3.24	–	2010–2011	Wang et al. (2013)	
Total	3.35	–	2010–2011	Wang et al. (2013)	
Suifen River population	Female	3.77	2.60	2012–2017	Wang et al. (2020)	
Male	2.85	1.91	2012–2017	Wang et al. (2020)	
Total	3.36	–	2012–2017	Wang et al. (2020)	
Tumen River population	Female	3.31	2.00	2020–2023	This study	
Male	3.14	1.87	2020–2023	This study	
Total	3.27	–	2020–2023	This study	

Fork length and body weight composition

The FL of Tumen River chum salmon ranged from 51.0 to 74.6 cm, with a mean of 61.8 ± 4.28 cm. The BW ranged from 1,400 to 5,200 g, with a mean of 2,516.95 ± 572.35 g. Most individuals (72.21%) measured 55 to 65 cm in FL. Females exhibited FL from 53.9 to 72.2 cm (mean 62.4 ± 3.74 cm) and BW from 1,600 to 4,020 g (mean 2,561.08 ± 518.14 g), with 74.69% falling within 55 to 65 cm in FL. Males showed FL from 51.0 to 74.6 cm (mean 60.4 ± 5.23 cm) and BW from 1,400 to 5,200 g (mean 2,398.78 ± 686.50 g), with 65.56% in the 55 to 65 cm FL range (Fig. 2, Table 3). Overall, these results demonstrated that females exhibit larger FL and BW than males.

Figure 2 Fork length (FL) frequency distribution of Tumen River chum salmon (5-cm intervals).

Most individuals (72.2%) have FL ranging from 55 to 65 cm, with 74.7% of females and 65.6% of males concentrated in this size range.

Table 3 Fork length (FL) and Body weight (BW) of age-2+ and age-3+ groups of Tumen River chum salmon (mean ± SD; sample sizes by sex and age group).

Age group	Sex	FL/cm	BW/g	n	
2+	Female	60.74 ± 3.44	2,388.38 ± 461.16	105	
Male	58.56 ± 4.69	2,165.49 ± 534.76	51	
Total	60.03 ± 4.02	2,315.51 ± 497.55	156	
3+	Female	63.33 ± 3.42	2,669.84 ± 509.39	125	
Male	62.48 ± 4.50	2,638.16 ± 618.98	38	
Total	63.13 ± 3.72	2,662.45 ± 537.11	163	
Notes.

Note: “+” indicates that the age is the annual ring value plus 0.7, for example, “2+” means that the age is 2.7.

Growth analysis

The ANCOVA results revealed no significant difference in the FL-BW relationship between sexes of Tumen River chum salmon (n = 331, P = 0.905 > 0.05). Therefore, the pooled data yielded the power function: BW = 0.0159×FL2.8969, where parameter b showed no significant different from 3 (t = 0.0336; Pauly, 1984) < t(0.05, 329), indicating isometric growth in chum salmon. Consequently, the VBGF was applied for modeling.

To investigate growth patterns across age groups, we fitted back-calculated FL (derived from scale radius) to age (Fig. 3). Figure 3 shows that the mean FLs at 2.7, 3.7, and 4.7 years were generally higher than those at 3, 4, and 5 years, suggesting distinct growth patterns among age groups (Wang, Liu & Tang, 2013). Due to limited samples in the 4+ and 5+ age groups, our analysis focused primarily on 2+ and 3+ age groups. Females exhibited greater FL and BW values across both age groups. Taking the 3+ age group as an example, ARSS analysis showed no significant growth difference between females and males (F = 0.3572 < F(0.01,2,5) = 13.2739). Consequently, sexes within the same age group were combined for analysis. The VBGF parameters (Table 4) and growth curves (Fig. 4) demonstrated that the 2+ age group grew faster than the 3+ age group, with FL growth rates declining with increasing age. ARSS further confirmed significant growth differences between the 2+ and 3+ groups. (F = 33.50 > F(0.01,1,4) = 21.198).

Figure 3 Back-calculated and measured fork lengths (FL) of (A) females and (B) males.

Measured FL at 2.7, 3.7, and 4.7 years old stages exceeded back-calculated FL at corresponding 3-year and 4-year stages, indicating divergent growth patterns among maturation cohorts.

Table 4 Growth parameters of age-2+ and age-3+ groups in different chum salmon populations (L∞, k, t0 as VBGF parameters; φ used to compare growth across age groups and populations).

Geographical population	Age group	L ∞	k	t 0	φ	Survey year	Source	
Ussuri River population	2+	114.198	0.247	−0.204	3.508	2010–2011	Wang et al. (2013)	
3+	90.300	0.294	−0.277	3.380	2010–2011	Wang et al. (2013)	
Suifen River population	♀2+	141.567	0.132	−1.245	3.423	2015–2017	Wang et al. (2019)	
♂2+	128.456	0.123	−1.557	3.307	2015–2017	Wang et al. (2019)	
♀3+	138.706	0.115	−1.560	3.345	2015–2017	Wang et al. (2019)	
♂3+	114.475	0.155	−1.134	3.307	2015–2017	Wang et al. (2019)	
Tumen River population	2+	141.574	0.135	−1.084	3.430	2020–2023	This study	
3+	135.66	0.13	−0.998	3.36	2020–2023	This study	
Notes.

Note: “+” indicates that the age is the annual ring value plus 0.7, for example, “2+” means that the age is 2.7.

Figure 4 Fork length growth and growth rate curves of 2+ and 3+ age groups.

Growth parameters L∞, k , t0: 141.574, 0.135, −1.084 for 2+ group; 135.66, 0.125, −0.998 for 3+ group. F-test revealed significant growth differences among age groups (F = 33.50 > F(0.01,1,4) = 21.198).

Fork length at first maturity

Based on the logistic formula (Fig. 5), the formulas for estimating the FL50% of female and male Tumen River chum salmon were derived as follows:

Figure 5 Logistic regression modeling of mature female and male chum salmon proportions across fork length classes.

Gray bands indicate 95% confidence intervals; step lines represent binned means at 2-cm intervals. Density ridgelines at “0” denote immature individuals per class, while those at “1” indicate mature individuals; intersections with the 50% maturity probability threshold delineate FL50% values. Parameters a, b: −27.708, 0.604 (females); −15.332, 0.349 (males); FL50% = 45.87 cm and 43.93 cm.

For females: Pi=11+e−−27.708+0.604li

For males: Pi=11+e−−15.332+0.349li.

The calculated FL50% for female and male were 45.87 cm and 43.93 cm, respectively, indicating that males attained sexual maturity earlier than females. ARSS analysis was applied to test the significance of differences in FL50% between sexes. The results demonstrated a highly significant divergence in sexual maturation probabilities between sexes (F = 14.385 > F(0.01,14,16) = 3.451).

Discussion

Age structure analysis

The determination of age is critical for sustainable fisheries management and evaluating the recovery of endangered populations. Sexual maturation age directly influences fish fecundity and longevity, making it a fundamental biological metric widely studied by researchers. Studies on chum salmon populations have revealed temporal fluctuations in their sexual maturation ages. For example, Helle & Hoffman (1995) documented an aging trend in sexual maturity among North American chum salmon populations, a pattern that has also been observed in Asian populations (Kaeriyama, 1996; Kaeriyama & Edpalina, 2008; Seo, Kudo & Kaeriyama, 2009). In contrast to the aforementioned conclusions, recent studies have documented that spawning populations of chum salmon returning to China’s Ussuri River, Suifen River, and Heilongjiang River exhibit a trend toward younger sexual maturation ages (i.e., earlier maturation timing) (Wang et al., 2013; Wang, Liu & Tang, 2013; Wang et al., 2020).

The Tumen River chum salmon population, as one of the four major geographic populations within China (Chen et al., 2005), has previously been mainly studied in terms of fecundity (Li et al., 2021), population conservation (Kang & Wan, 2016; Wang et al., 2023), and genetic structure (Han et al., 2020; Yan et al., 2021). The absence of historical age data has led this study to focus solely on the current age composition of the population. Based on the fact that individuals aged 2+ and 3+ account for 96.37% of the total, this indicates that the current age structure is dominated by young individuals. Furthermore, the Ussuri River and Suifen River populations, used for comparing maturation age with the Tumen River population in this study, had their average maturation ages reported by Wang et al. (2013) and Wang et al. (2020) as historic lows during the study period. Therefore, the lower mean age of the Tumen River population and the lower T50% values of both its female and male groups indicate that the current maturation age exhibits a young profile.

The interaction between the genetically influenced sexual maturation rate and abundance differences across adjacent generations jointly drives shifts in the age structure of spawning populations (Kaev, 2000). That is to say, the young age structure of the Tumen River chum salmon population is attributed not only to the high abundance of low-age fish returning to spawn but also potentially to the earlier sexual maturation of faster-growing individuals. This is corroborated by the number of returning groups at age 2+ and 3+, as well as the earlier maturation timing and faster growth rate of the 2+ age group compared to the 3+ age group observed in this study. Beyond genetic influences, anthropogenic factors such as overfishing may contribute to earlier maturation (Fukuwaka & Morita, 2008; Hard et al., 2008). Historical records show that the Tumen River chum salmon catch declined from over 100,000 individuals in the early 20th century to less than 1/10 of the 1940s level in recent years (Kang & Wan, 2016; Li et al., 2021). Overexploitation likely exerts selective pressure, prompting populations to mature earlier to ensure reproductive continuity.

Growth characterization

The Tumen River chum salmon population exhibited BW of 1,400–5,200 g and FL of 51.0–74.6 cm. Individuals with FL of 60–65 cm accounted for the largest proportion (45.52%), while those <60 cm and >65 cm constituted 31.72% and 22.66%, respectively. This distribution indicates a predominance of small-sized individuals in the population. The FL-BW relationship was described as BW=0.0159 × FL2.8969, where the regression coefficient b showed no significant difference from 3, indicating isometric growth and suitability for simulation with VBGF, consistent with Ussuri and Suifen River populations (Wang et al., 2013; Wang et al., 2020). ARSS analysis revealed significant growth differences between the 2+ and 3+ age groups. The 2+ group displayed higher growth coefficient (k) and performance index (φ) values (Table 4), corresponding to faster FL growth rates and elevated growth curves (Fig. 4). This confirms that earlier-maturing individuals grow faster, with maturation age inversely related to FL growth rate, consistent with Morita et al. (2005). Similar to most fish, rapid growth in younger individuals enhances population survival (Duffy & Beauchamp, 2011). Earlier-maturing individuals exhibit greater environmental adaptability, but increased abundances intensify competition for food resources, suppressing growth in older individuals and progressively slowing their growth rates. Using φ, we further compared chum salmon growth rates among populations in the 2+ and 3+ age groups (Table 4). The results indicate that the Tumen River population exhibits an intermediate rate—faster than the Suifen River population but slower than the Ussuri River population.

Sexual maturity in fish is influenced by both age and body size. Given FL variability at maturity, we used FL50% to represent Tumen River chum salmon maturation size, reflecting population growth and fecundity to prepare for assessing of reproductive potential and fisheries management. The FL50% values were 45.87 cm for females and 43.93 cm for males, and the corresponding ages (T50%) were 2.00 and 1.87 years, respectively, demonstrating earlier male maturation at smaller sizes. Females’ larger size at maturity aligns with natural selection pressures, as their reproductive success strongly correlates with body size (Wang et al., 2020; Li et al., 2021), while prolonged energy allocation for gonadal development further necessitates delayed maturation. Comparatively, Ussuri River populations exhibit FL50% values of 48.36 cm (females) and 46.60 cm (males), while Suifen River populations show 51.53 cm (females) and 42.15 cm (males) (Wang et al., 2013; Wang et al., 2020). Tumen River females display smaller FL50% values than both populations, whereas males exhibit marginally larger FL50% than Suifen River males but are close in value. This indicates that the current Tumen River population exhibits characteristics of small body size at sexual maturity. Chum salmon exhibit high growth plasticity due to genetic and environmental influences. As an anadromous species, they demonstrate remarkable homing precision to natal spawning grounds, resulting in significant morphological and ecological divergence among geographic populations (Saito, 2017; Saito et al., 2020). After hatching, fry spend minimal time in freshwater before migrating to marine environments where they undergo most growth and development. Consequently, their body size is highly sensitive to marine temperature (Farley & Moss, 2009; Kasugai et al., 2012), climate change (Kitada & Kishino, 2019; Kuroda et al., 2020), and interspecific/intraspecific competition (Ishida et al., 1993; Helle & Hoffman, 1998; Bigler, Welch & Helle, 1996; Tadokoro et al., 1996). Given the Tumen River chum salmon’s significant scientific and economic value, small body sizes not only directly reduce fecundity and impair population recovery capacity, diminish marine-to-terrestrial nutrient transport, but also indirectly lead to decreased meat yield, increased processing costs, and lowered market prices, ultimately affecting fishermen’s income and regional economic stability. Therefore, minimum harvest size limits should exceed FL50%, with sustained monitoring of this metric. Quantitative management of minimum sizes is critical for mitigating the miniaturization of returning offspring cohorts.

Conclusion

The Tumen River chum salmon population primarily consists of 2+ and 3+ age groups, with different growth patterns among age groups. Among them, the group that reaches sexual maturity earlier exhibits a faster growth rate. In addition, most individuals in the population have a fork length of less than 65 cm, indicating that the current returning population of Tumen River chum salmon is mainly composed of young-aged and small-sized individuals. These findings provide fundamental data for understanding the biological characteristics of the Tumen River chum salmon population and offer critical data support for subsequent resource management and conservation initiatives.

Supplemental Information

Supplemental Information 1 Raw data on age and growth of chum salmon from the Tumen River

Additional Information and Declarations

Competing Interests

Author Contributions

Animal Ethics

Data Availability

The authors declare there are no competing interests.

Anqi Wang conceived and designed the experiments, performed the experiments, analyzed the data, prepared figures and/or tables, authored or reviewed drafts of the article, and approved the final draft.

Peilun Li conceived and designed the experiments, performed the experiments, authored or reviewed drafts of the article, and approved the final draft.

Fujiang Tang conceived and designed the experiments, authored or reviewed drafts of the article, and approved the final draft.

Shuhan Xiong analyzed the data, prepared figures and/or tables, authored or reviewed drafts of the article, and approved the final draft.

Jiaoyang Yu analyzed the data, authored or reviewed drafts of the article, and approved the final draft.

Jilong Wang conceived and designed the experiments, authored or reviewed drafts of the article, and approved the final draft.

The following information was supplied relating to ethical approvals (i.e., approving body and any reference numbers):

Heilongjiang River Fisheries Research Institute provided full approval (No.: 20200920-001) for this research.

The following information was supplied regarding data availability:

Raw data is available in the Supplemental Files.

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
