# Peer review of "Age and growth of chum salmon from the Tumen River"

_PeerJ, doi:10.7717/peerj.19902_

## Round 0.1 · original submission · Major Revisions

According to the reviewer comments, it has been decided to evaluate your article as "Major Revision". I am sending your manuscript back to you for you to edit it again according to the reviewer comments. Please review the reviewer comments carefully.

**Language Note:** The review process has identified that the English language must be improved. PeerJ can provide language editing services - please contact us at [email protected] for pricing (be sure to provide your manuscript number and title). Alternatively, you should make your own arrangements to improve the language quality and provide details in your response letter. – PeerJ Staff

Reviewer 1 ·

Basic reporting

There were a number of English language grammatical errors throughout the manuscript, but they were primarily minor errors, and sentences were largely understandable. The writing overall was simple, repetitive, and of low quality. Overall, there was not enough background or explanatory information provided to adequately understand some of the analyses and why they were performed. Often, parameters were not defined or were defined poorly. Information provided in the methods was not sufficient. The majority of my review is in section 4 where I have comments for the different sections as well as line items.

Experimental design

Seemed to follow the analysis of Wang et al. 2013 and 2019 which I was unfortunately unable to locate. This was a basic description of the age structure and growth of the Tumen River Chum salmon population. I did not understand the purpose of the logistic regression analysis related to sexual maturity.

Validity of the findings

The reporting of the age structure and weight at capture was interesting and important; however, the age notation needed to be more clearly defined in the methods.

Overall I was pretty confused with the rest of the analyses performed. A large component of the analysis was estimating FL50%, whose definition and biological relevance were not clearly communicated. From what I could tell, it meant the size at which there is a 50% probability that a fish will be sexually mature. However, Chum salmon spawn a single time in their life right before they die. This is when they are sexually mature, and it was also when they were captured for this study. Table 2 reports average fork lengths of age 2+ and 3+ fish from this study, with sizes ranging from about 58cm to 63cm. However, FL50% is estimated to be 48.87cm for females and 43.93cm for males. Why is FL50% so much smaller than the actual average FLs of sexually mature fish returning to spawn?

This is one of a number of issues. Please see section 4 below for details.

Additional comments

Abstract:
Line 17: “…were examined for scales and basic biological traits…:” Rewrite this sentence for clarity.

Line 20: ARSS indicated no sexual difference in what? Growth? State clearly what was being evaluated.

Line 22: “…Tumen River chum salmon are currently severely underaged.” This statement makes it sound like the methods used to estimate fish age are incorrectly assigning ages to fish that are less than their true ages. I think you mean that this population has a young age structure.

Line 24-25: What different chum populations? So far, you have only indicated the Tumen River population.

Introduction:
The introduction is extremely short and repetitive. It provides almost no background information on chum salmon or on the study population beyond saying that chum are anadromous, this is one of four major chum populations in China, and it has been overfished. Is this a hatchery or wild population? Is there any information on run size? Scales seem to be an integral part of this study, but they are never even mentioned in the introduction. An explanation of how scales grow and why they are useful for an age and growth study would be valuable. Research objectives are poorly described.

Line 33-35: Reference for this?
Line 42: How does analyzing biological traits enrich the biology of chum salmon? Too general.

Methods:
Line 52: Does this mean this is a hatchery population? Is there also a wild population in the river? No mention of this in the introduction.

Line 61: I was unable to find Wang et al. 2012 to evaluate the scale ageing and measurement protocol. It would be worth including a summary of what scale process followed in the methods section.

Line 62: Please state clearly what the purpose of this equation is; “calculating the chum salmon fork length” is not sufficient. Also, provide a reference if it came from a previous study. It appears to be a fork length back calculation formula based on fish and scale size, with juvenile parameters taken from Fukuwaka and Morita 1994. Also, please indicate why you needed a back calculation formula.

Line 89: What is a whorl? What constitutes the last whorl formation? I assume this is referring to scales but scales are not even mentioned in this section. Neither “scales” nor “whorls” are mentioned in Silverstein et al. 1998, which is referenced in this sentence. Additionally, Silverstein et al is focused on Chinook salmon, not chum salmon. If this is what is being used to estimate the time of sexual maturity, it needs to be explained in the methods and have a relevant citation.

Lines 91-94: I’m pretty confused as to how and why this logistic regression was applied to estimate FL at sexual maturity. Chum salmon become sexually mature at the time of their freshwater migration, which is when they are captured in the gill nets for this study. The true FL at sexual maturity is already known.

Results:
What exactly is FL50%, and why is it biologically important? The explanation in this section on line 88 states “the 50% first sexually mature fork lengths (FL50%).” Does that mean the size at which there is a 50% probability that a fish will be sexually mature? In Table 2, you report the average fork lengths of age 2+ and 3+ fish from this study with sizes ranging from around 58cm to 63cm. These are fish that were captured in gill nets on their spawning migration, so they would be considered sexually mature individuals. However, in the abstract you report FL50% to be 48.87cm for females and 43.93cm for males. Why is FL50% so much smaller than the actual average FLs of sexually mature fish returning to spawn?

Line 97: What does an age “2+” fish mean? I am assuming you used this notation so that your results could be compared to the referenced Wang et al. papers that use the same notation. However, this age notation needs to be explained in the methods.

Line 111: Some fish you assign ages of 2.7, 3.7, and 4.7, but others you are assigning ages of 3 and 4? Which fish were given whole number ages and why?

Discussion
Line 136: Instead of “Foreign” say “Studies from outside China…” or something to that effect.

Line 141: Remove “have a relatively simple age composition”. All Chum salmon populations have relatively simple age structures.

Line 145: When comparing your study to data from other studies, include the citation.

Line 147: Are you speculating this? If someone else speculated this, what is the source?

Line 148: This information on run size is important and should be brought up in the introduction.

Line 160: Should be “Duffy and Beauchamp 2011”

Line 161: Please rewrite this sentence to clarify the meaning.

Line 175: Chum salmon only spawn once in their lifetime, so saying their “Fork length at first sexual maturity” is unnecessary. You caught them on their way to spawn and know their true size at sexual maturity.

Tables:
Tables should be stand-alone, meaning the symbols used should be defined in the captions, and units should be provided.

Table 2 – No units provided for FL or weight.

Table 3 – Define symbols in caption. Include units when applicable.

Table 4 – I recommend writing out the population names in the table. T50% is introduced in this table. It is not mentioned in the methods, the results, or the discussion. Chum salmon spawn once in their life and then die within a week or so. Their age at that time should be considered their age at sexual maturity. I do not understand how T50%, described in the table as the “Age of sexual maturity” for a population, can be more than a year less than the mean age of fish in a population.

Table 5 - Define symbols in caption. Include units when applicable.

Figures:
The first figure for this paper should be a map to orient the reader as to where the study population is located. Because scales were a large component of the back calculation and ageing portion of this study, an example scale image could be another useful figure.

Cite this review as

Reviewer 2 ·

Basic reporting

In this study, the authors investigated the age structure and growth characteristics of the Tumen River chum salmon. 331 individuals caught between 2020 and 2023 were examined for scales and basic biological characteristics and divided into four age groups. The von Bertalanffy growth equation was used to model growth among different age groups and sexes, and the parameters were estimated by maximum likelihood. Using residual sum of squares (ARSS) analysis, the authors found that there were no differences between sexes, but significant variations between age groups 2+ and 3+, and logistic regression showed that the size at maturity (FL%50) was 45.87 cm for females and 43.93 cm for males, and ARSS showed significant sexual differences at FL%50.

Experimental design

The authors performed age determination using body scales of Chum salmon sampled with three-layer drift gill nets (mesh size 12.67 cm) from the Tumen River between mid-October 2020 and mid-2023. The relationship between fork length and weight, von Bertalanffy growth constants, and 50% first sexual maturity lengths were analyzed in accordance with the literature.

Validity of the findings

The authors have obtained significant findings from the growth and maturity parameters they analyzed.
Conclusions are well-articulated, related to the original research question, and supported by the results.

Cite this review as

Reviewer 3 ·

Basic reporting

The English language should be improved. While your content is mostly comprehensible, some of the phrasing and grammar is awkward or incorrect. For example, in the introduction, chum salmon are referred to in the singular, i.e. “Chum salmon is a typical anadromous fish that reproduces only once in its lifetime…”
Instead of singular, use the plural to refer to chum salmon, i.e. “Chum salmon are a typical anadromous fish that only reproduce once, and are widely distributed in the North Pacific Ocean…”
The entire article would benefit from rewriting with chum salmon referred to in the plural instead of singular.
English can be improved, and writing made more concise in the sample collection section. I recommend changing lines 51-55 to:
The sampled chum salmon were caught in the Hunchun section of the mainstream Tumen River in mid-October of 2020-2023 with triple layer drift gillnets (mesh size 12.67 cm) for use as broodstock. Biological measurements were taken during the artificial propagation process, and 6-10 dorsal scales were taken above the lateral line.

Line 133 change to “Usually, the age of maturity in fish is mainly affected by genetic and environmental factors.”
Line 136 change its to it is

The raw data files should also include the age of each fish, not just the length and weight.

Experimental design

Section 2.2.1:
This section could be clearer. Unclear what LT and ST are referring to, as line 65 just says they are “fork length and scale diameter at the age.” What age? I assume that ST is the scale diameter measured on the scale at the annuli of interest, but that could be clarified.
Line 89: Circuli or annuli more typically used instead of whorl
2.2.3
Since all of the captured fish were returning to fresh water to spawn, presumably they were all sexually mature? I am unclear on the use of the last whorl formation in chum salmon as the basis for the delineation of their sexual maturity. The referenced paper did not provide any additional clarity.

Validity of the findings

Line 136-137: Studies in Alaska, particularly in Chinook, have shown a declining trend, where age at maturity is generally decreasing, in contrast to the statement by the authors that "Foreign studies show an increasing trend in the sexual-maturity age of Pacific salmon in North America..."

For example, the percentage of Chinook salmon maturing at age 6+ in the Yukon River and Nushagak River dropped by half since the early 1980s.
See:
Brown, R. J., Bradley, C. and Melegari, J. L. 2020. Population trends for Chinook and summer Chum salmon in two Yukon River Tributaries in Alaska. Journal of Fish and Wildlife Management, 11(2):377–400.
Ruggerone, G. T, B. M. Connors, B. A. Agler, L. I. Wilson, and D. C. Gwinn. 2016. 2016 Arctic Yukon Kuskokwim Sustainable Salmon Initiative Project Product.

Line 172: “Differences in marine habitats…also contribute to growth disparities among different chum salmon geographic populations.” Has there been work to determine if the chum populations from the different rivers occupy different marine areas? Is it possible they intermix in the marine environment and then return to their natal rivers?

Additional comments

Interesting data on a little studied population of chum salmon, however the article could overall benefit from improvement of the English language, and additional clarity on the methods and outcomes.

Cite this review as

---

## Round 0.2 · Minor Revisions

Thank you for the corrections you have made. However, our reviewers would like some further corrections to be made. Please carefully review the corrections that have been reported.

Reviewer 3 ·

Basic reporting

Line 17-18 change “through scale observations and biological measurements” to “using scale growth and biological measurements.”
Abstract says ARSS indicated no significant growth differences between sexes but then 4 lines later says that ARSS revealed highly significant sexual differences in FL50%.
Line 23-27: Could be clearer that the relationship between bodyweight and fork length was not significantly different between the sexes, but there were significant differences in fork length by sex.
Line 27-31: The conclusion in the abstract of “These finding suggest a younger age structure and smaller body size in the current Tumen River population…” in comparison to what? The historical Tumen River population? Other chum throughout the world? The following sentence indicates that other nearby populations in the Ussuri and Suifen rivers have similar age structures.
Line 37: Recommend changing to- “Chum salmon (Onchorhynchus keta) are semelparous and anadromous. They spawn in freshwater, and fry outmigrate to the marine environment shortly after emergence. Maturing adults return to their natal streams to spawn at various ages, typically between two and five years. All die after spawning (Salo 1991).”
1st paragraph, lines 37-49: change “it” to “they” when referring to chum salmon. Ie, line 41, “in China, they are restricted…”
Line 42: change to “Historically, Tumen River annual chum salmon catches exceeded 100,000, but spawning ground degradation caused by hydropower development and environmental pollution, combined with overexplotation, reduced annual catches to 2,000-3,000.”
Paragraph lines 50-66: The English phrasing and word usage reads oddly in this paragraph. Recommend having a native English speaker review. The authors change between circuli and circuit. The first sentence in this paragraph (line 50-51) is odd and out of place. Suggested edits:
Fish biology studies, establish the premise for resource conservation, with age and growthanalyses central to biological investigations. Chum salmon scales provide critical data for age determination and growth pattern analysis through their circuli structurepatterns of growth. As scales grow proportionally with somatic development, seasonal metabolic shifts drive distinct variations in circuli patterns (annuli formation)(Hile, 1936; Shen, 2011; Wang et al., 2012). , During rapid spring and summer growth, circuli are widely spaced, producing widely spaced circuli that form sparse bands during rapid spring/summer growth, while during slow winter growth, circuli are more densely spaced. densely packed circuli create compact bands during winter growth cessation. These patterns of summer and winter growth alternating annuli serve as reliable annual markers (98-99% inter-observer consistency; Morita and Fukuwaka, 2006) for age determination (98-99% inter-observer consistency; Morita and Fukuwaka, 2006).
What is the relevance of including the below information (lines 57-66)?
Juvenile chum salmon are approximately 40-50 mm FL Additionally, when scales form, the fork length of chum salmon is approximately 40-50 mm (Kaeriyama, 1989; Shimoda et al., 2019). Based on this, Irie (1990) and Mayama and Ishida (2003) noted that the fork lengths at the formation of the 5th and 10th circuit on scales are approximately 60-80 mm and 90-100 mm respectively, which establishes 70-80 mm fork length as the threshold for initiating offshore migration. Furthermore, Saito et al. (2011) measured the distance from the scale center to the 5th circuit (regarded as the coastal habitat phase) and the distance between the 5th and 10th circuits (regarded as the offshore migration phase) to investigate growth rates under different environmental conditions. Wang et al.(2013a; 2020) derived growth variations at different life stages by measuring distances from the scale center to annuli, exploring their relationships with environmental factors. This collectively demonstrates the critical role of scale analysis in studying chum salmon age and growth, providing methodological support for subsequent resource assessments.
Lines 71-81: The authors state that in China, research on chum salmon age and growth has been short term with long intervals between studies, then state that studies on Ussuri River chum have spanned from 1952 to 2011. That sure seems long term to me- so if they mean that there were short studies in 1952 then a short study in 2011 (short study, long interval), then they should state that more clearly.
But given that the next sentence compares this to a brief mention of the Tumen River population by Li et al, it appears that the Ussuri R population has been well studied.
Line 77: analyzing fork length based on scale morphology doesn’t really make sense. Do the authors mean “(1) Analyze biological characteristics, including age and size, using scale growth patterns and body measurements”?
Line 79: remove patterns, just say “identify variations in age at maturity and fork length;”

Experimental design

Line 86-87: Unclear what is meant by “breeding and releasing station for temporary holding.” Are fish released to die in the river after being stripped of eggs and milt?
Section 2.2: All discussion of scales here uses annuli and annulus, where the introductions talks about circuli and circuits.
Section 2.2.5: While I appreciate the clarification the authors added on the use of “fork length at first maturity,” I still find it to be a somewhat confusing term, since chum salmon only reach sexual maturity once in their lives. Perhaps just call it fork length at maturity? FL50% makes sense as the FL at which 50% of the population reach sexual maturity. Remove the mentions of first maturity, and I think this makes sense. Same thing with removing first from the description of T50%. Because again, this is not the age at which the first sexual maturity is observed within the population, but the age at which 50% of the population achieve maturity in a given year.
Line 140-141: what critical maturation phases, how are they synchronized to annulus formation.
Line 144: you use age 3+ and age 4 to describe fish ages here, although you state earlier you did not use the calendar year method for this study.

Validity of the findings

Line 155: what percentage of males were age 2+? Seems odd to me that authors state that females were predominantly 3+ age, and males were mainly 2+, but the mean and T50% values for both sexes were 3+ and 2+.
Lines 162-167: For the way these sentences are structured, it doesn’t make sense to have the FL and BW in parentheses.
Line 163-64: The dominant FL group is not explained- is this a length frequency? If so, 57.5-67.5 is an odd size bin. Why not do length frequency for 5 cm or 10 cm increments? Same comment for the males and females separately.
Line 175: I don’t think that having a high FL at 3.7 years old vs 3 years old is a distinct growth mode. We would expect older fish to keep growing, yes?
Line 177-178: The authors say results imply that females have faster growth. Then the next sentence says that ARSS revealed no sexual differences in 3+ growth, so sexes were combined. These sentences appear contradictory- are there sexual differences in growth or not?
Line 208-209: “…leaving sexual maturation age trends undetermined.” This study does not determine sexual maturation age trends either, as it only looks at the current sexual maturation age.
Line 210-211: This sentence is unclear, and seems to state that the Ussuri and Suifen river populations had the lowest recorded sexual maturation age, but the previous sentence states that the Tumen has a younger age structure.
Line 225: Again, why the odd interval of 61.5-63.5 cm? And a different size of interval than previous. Also , the median is the value separating the lower half from the upper half, so if 45% are below and 34% are above, that’s not the median value.
Lines 236-238: Tumen R salmon grow faster than Suifen fish but slower than Ussuri fish, but mature at a younger age than both populations? Which is somewhat contradictory to the previous statement that maturation age is inversely related to FL growth rate.
Line 239-40: What do the authors mean by some individuals maturing before the breeding season? Are they implying that some salmon are ready to reproduce before their usual spawn timing, or is this just a general statement about fish?
Line 242-243: Male maturation at smaller sizes does not necessarily mean earlier male maturation in terms of age.
Line 249: Again, I’d be wary of using words like trend when talking about this single study, not with multiple years analyzed individually over time.
Line 258: Again, this paper does not provide evidence that the Tumen R salmon body size is declining, only that the fish analyzed in this paper are generally smaller than their cohorts in other rivers.
Conclusion: Interesting that there are different growth patterns among age cohorts, and that the authors demonstrate that males mature earlier than females. But yet there is no growth difference between females and males.
And again, saying there is a miniaturization trend in Tumen River fish without previous size and age compilations to compare to is not correct.

Additional comments

Figure 1: The caption says “spatial distribution of chum salmon sampling sites…” but there is only one site notated on the map. Either correct the caption to read “Location of chum salmon sampling site” or add the actual sampling sites (if there were multiple).
Figure 2: The intervals of 2.5 cm, and starting at 53 cm, read a little oddly. Perhaps use mm, in which case 25 mm intervals make a little more sense? But again, starting at 53.0 makes for an odd appearance.
Figure 3. I am confused by this- the authors calculated fork length based on scale size, but found that actual fork lengths exceeded the calculated lengths based on scales? So wouldn’t that imply that all lengths calculated using scales would be inaccurate, since the ground truthing did not find consistency?

Cite this review as

---

## Round 0.3 · accepted · Accept

Thank you for your corrections. After the changes, your article has been accepted.